# Fine-grained Conversational Decoding via Isotropic and Proximal Search

**Yuxuan Yao**   **Han Wu**   **Qiling Xu**   **Linqi Song**[*]
City University of Hong Kong
yuxuanyao3-c@my.cityu.edu.hk   hanwu32-c@my.cityu.edu.hk
qilingxu2-c@my.cityu.edu.hk   linqi.song@cityu.edu.hk

## Abstract

General-purpose text decoding approaches are usually adopted for dialogue response generation. Although the quality of the generated responses can be improved with dialogue-specific encoding methods, conversational decoding methods are still under-explored. Inspired by Wu et al. (2023) that a good dialogue feature space should follow the rules of locality and isotropy, we present a fine-grained conversational decoding method, termed *isotropic and proximal search (IPS)*. Our method is designed to generate the semantic-concentrated response, while still maintaining informativeness and discrimination against the context. Experiments show that our approach outperforms existing decoding strategies in the dialogue field across both automatic and human evaluation metrics. More in-depth analyses further confirm the effectiveness of our approach.

## 1 Introduction

Dialogue response generation (Li et al., 2017; Wang et al., 2020) aims to generate the utterance that forms a coherent and fluent continuation given a dialogue context. Generic text decoding strategies (Rieser et al., 2014; Ritter et al., 2011; Chen et al., 2017) are usually adopted to produce grammatical and contextual responses. As an independent technique, decoding strategy can also enhance the generation quality of large language models.

Existing text decoding methods have been explored in various generic text generation tasks, but lack tailoring for dialogue generation, e.g., capturing dialogue-specific features and generating an informative and discriminative dialogue response (Su et al., 2021; Wu et al., 2023). Early maximization-based methods, e.g., greedy search (Li et al., 2016b) and beam search (Wiseman et al., 2017), may lead to dullness and degeneration (Fan et al., 2018; Holtzman et al., 2018). Later sampling-based improvements are proposed to tackle these problems, including top-$k$ sampling (Fan et al., 2018) and nucleus search (Holtzman et al., 2018). While alleviating degeneration, these sampling methods introduce critical semantic inconsistency and are not aligned with human-written prefix (Basu et al., 2021). Specifically, a bunch of studies (Ethayarajh, 2019; Su and Collier, 2023) have asserted that the problem of *anisotropy*, i.e., a distribution pattern in the latent space with features occupying a narrow cone in the space, leads to inconsistency and degradation of the generation. Although contrastive search (Su et al., 2022) has been proposed correspondingly to mitigate the issue, as a generalized text decoding strategy, it still ignores dialogue-specific features, such as utterance dependencies and conversational structure information. Therefore, research on conversational decoding methods is warmly needed.

In this work, we propose a fine-grained conversational decoding method, namely **i**sotropic and **p**roximal **s**earch (IPS). Different from traditional approaches, we consider the previous tokens and contexts separately from a granular perspective. Acknowledging that locality and isotropy are two important properties for refining the dialogue feature space, we design our IPS following these rules: (i) the generated output should be selected from the most probable candidate set predicted by the dialogue model; (ii) the generated tokens in the same utterance should be proximal to each other for expressing a concentrated idea; and (iii) the newly generated utterance should be discriminative enough with respect to the context utterances. In this way, our method encourages informativeness and discrimination among different utterances as well as maintains a concentrated idea within an utterance. We evaluate our approach on two commonly-used dialogue datasets, DailyDialog (Li et al., 2017) in English and LCCC (Wang et al., 2020) in Chinese. Both human and automatic evaluation results, i.e., indicators based on GPT3.5,

---

[*]Corresponding author: linqi.song@cityu.edu.hk

consistently show that IPS can generate more fluent, coherent, and human-like responses than existing decoding methods.

## 2 Methodology

### 2.1 Preliminary

**Dialogue response generation**   Given a dialogue context $D = \{u_1, u_2, ..., u_N\}$ composed of $N$ utterances, where $u_i = \{x_{i,1}, x_{i,2}, ..., x_{i,|u_i|}\}$ is a sequence of consecutive words, the task of dialogue response generation is to produce the continuation utterance $u_r = \{w_1, w_2, ..., w_{|u_r|}\}, (r = N + 1)$.

There are generally two key steps to finish the task, including context encoding and response decoding. For the first step, we obtain the context representations $\mathbf{H}$ from the language model by concatenating the utterances into a sequence.

$$\mathbf{H} = \text{PrLM}(u_1 \, [\text{EOU}] \, u_2 \, [\text{EOU}] \, ... \, u_N \, [\text{EOU}]),$$

where [EOU] is the special token inserted as the last token of each utterance.

For the decoding step, the response is generally produced in an auto-regressive manner as follows

$$p(w_{1:|u_r|}) = \prod_{i=1}^{|u_r|} p(w_i|w_{<i}, D) \qquad (1)$$

**Dialogue modeling**   Wu et al. (2023) has demonstrated that locality and isotropy are two key properties for building a good conversational feature space. Specifically, locality encourages the model to aggregate the representations of tokens within an utterance while isotropy pushes away the representations of distinct utterances.

### 2.2 Isotropic and Proximal Search

We present a fine-grained conversational decoding method, i.e., isotropic and proximal search (IPS). Specifically, we expect the generated response to satisfy two requirements: 1) representations of the response tokens are nearby to convey a concentrated idea, saying proximity; 2) the response representation is discriminative to the context utterance representations, saying isotropy.

During the decoding stage, for proximal search, we try to select the candidate token having the shortest average distance to the existing generated tokens. For isotropic search, we try to choose the token that enables the response representation most discriminative to representations of context utterances. As the response representation cannot be

determined during the decoding stage, we calculate it in an approximate way, i.e., averaging the representations of the already generated tokens, as follows:

$$\mathbf{h}_{RT} = \frac{1}{T} \sum_{i=1}^{T} \mathbf{h}_{w_i} \qquad (2)$$

where $\mathbf{h}_{RT}$ is the response representation which will be dynamically updated along with the generation process, and $T$ is the number of already generated tokens.

Up to now, the problem changes to how to generate the first token for starting the isotropic and proximal search since the method is heavily dependent on the previous tokens. To address this problem, we attempt to finish the first $n$-steps generation by traditional decoding methods, such as beam search, top-$k$ sampling or nucleus sampling. On the other hand, as IPS is essentially a deterministic decoding strategy, this solution also enables it to produce diverse responses by using different decoding strategies in the first $n$ steps. Therefore, in each step $t$ after the first sampling stage, we calculate the proximal and isotropic values as follows:

$$\text{p\_value}_t = \frac{1}{t - 1} \sum_{i=1}^{t-1} s(\mathbf{h}_{w_t}, \mathbf{h}_{w_i}) \qquad (3)$$

$$\text{i\_value}_t = \frac{1}{N} \sum_{i=1}^{N} s(\mathbf{h}_{RT}, \mathbf{h}_{u_i}) \qquad (4)$$

where $s$ is the cosine similarity. $\mathbf{h}_{u_i}$ are the utterance representations obtained from the special token [EOU]. The proximal value measures the average distance between the candidate token and the already generated tokens while the isotropic value stands for the average similarity between the undergoing response representation and all utterance representations. Next, the selection of the candidate token $w_t$ is formulated as,

$$
\begin{aligned}
w_t = \underset{w_t \in V^{(m)}}{\text{argmax}} \{ \alpha \times \underbrace{p(w_t \mid w_{<t}, D)}_{\text{model confidence}} \\
+ (1 - \alpha) \times \underbrace{(\text{p\_value}_t - \text{i\_value}_t)}_{\text{isotropic and proximal penalty}} \}
\end{aligned}
\qquad (5)
$$

where $V^{(m)}$ is the set of top-$m$ predictions from the model's probability distribution $p(w_t \mid w_{<t}, D)$ and $m$, is typically set as $4 \sim 8$. In Eq. (5), the first term, model confidence, is the probability of the candidate $w_t$ predicted by the model. The second term, isotropic and proximal penalty, aims to maximize the discrimination between the undergoing response and previous utterances and minimize the

| Model | Strategy | DailyDialog | | | Distinct | | LCCC | | | Distinct | |
|---|---|---|---|---|---|---|---|---|---|---|---|
| | | BS ↑ | MV ↑ | GE ↑ | Dis2 ↑ | Dis4 ↑ | BS ↑ | MV ↑ | GE ↑ | Dis2 ↑ | Dis4 ↑ |
| BART | greedy | 0.1275 | 0.569 | 2.17 | 0.344 | 0.776 | 0.0636 | 0.062 | 1.88 | 0.126 | 0.437 |
| | beam | 0.1317 | 0.599 | 2.29 | 0.341 | 0.755 | 0.0639 | 0.145 | 1.91 | 0.155 | 0.466 |
| | top-$k$ | 0.1312 | 0.623 | 2.20 | 0.350 | 0.780 | 0.0648 | 0.154 | 1.94 | 0.152 | 0.487 |
| | nucleus | 0.1298 | 0.642 | 2.34 | 0.352 | 0.791 | 0.0626 | 0.178 | 1.91 | 0.156 | 0.534 |
| | contrastive | 0.1147 | 0.622 | 2.07 | **0.396** | **0.810** | 0.0538 | 0.205 | 1.90 | **0.190** | **0.583** |
| | IPS | **0.1335** | **0.647** | **2.43** | 0.355 | 0.798 | **0.0653** | **0.212** | **1.98** | 0.176 | 0.540 |
| SimCTG ($\rho = 0.5$) | greedy | 0.1099 | 0.447 | 2.21 | 0.306 | 0.709 | 0.0678 | 0.088 | 1.82 | 0.137 | 0.470 |
| | beam | 0.1196 | 0.556 | 2.27 | 0.314 | 0.713 | 0.0692 | 0.206 | 2.02 | 0.179 | 0.539 |
| | top-$k$ | 0.1169 | 0.544 | 2.06 | 0.322 | 0.733 | 0.0695 | 0.195 | 2.11 | 0.168 | 0.534 |
| | nucleus | 0.1169 | 0.571 | 2.32 | 0.327 | 0.753 | 0.0680 | 0.223 | 2.10 | 0.169 | 0.575 |
| | contrastive | 0.1123 | 0.608 | 2.17 | **0.395** | **0.807** | 0.0607 | 0.278 | 1.98 | **0.197** | **0.618** |
| | IPS | **0.1293** | **0.628** | **2.36** | 0.359 | 0.787 | **0.0704** | **0.294** | **2.31** | 0.196 | 0.580 |
| SimDRC ($\delta = 0.7$, $\alpha = 0.3$) | greedy | 0.1255 | 0.560 | 2.06 | 0.345 | 0.774 | 0.0699 | 0.090 | 2.21 | 0.136 | 0.471 |
| | beam | 0.1315 | 0.632 | 2.18 | 0.338 | 0.745 | 0.0715 | 0.196 | 2.11 | 0.180 | 0.543 |
| | top-$k$ | 0.1068 | 0.648 | 2.20 | 0.345 | 0.773 | 0.0720 | 0.203 | 2.19 | 0.166 | 0.540 |
| | nucleus | 0.1284 | 0.632 | 2.16 | 0.353 | 0.793 | 0.0697 | 0.226 | 1.88 | 0.166 | 0.569 |
| | contrastive | 0.1174 | 0.653 | 2.16 | **0.397** | **0.819** | 0.0613 | 0.271 | 2.21 | **0.197** | **0.614** |
| | IPS | **0.1336** | **0.665** | **2.46** | 0.366 | 0.800 | **0.0722** | 0.272 | **2.32** | 0.192 | 0.569 |

Table 1: Automatic evaluation results on DailyDialog and LCCC, where BS means F1 value of BERTScore (Zhang* et al., 2020), MV represents MAUVE (Pillutla et al., 2021), and GE represents G-Eval (Liu et al., 2023).

token difference within the response. The hyper-parameter $\alpha \in [0, 1]$ regulates the importance of these two components. When $\alpha = 1$, our method degenerates to the greedy search method.

We claim our method is *fine-grained* because the generic auto-regressive generation predicts the next token by jointly considering the already generated tokens $w_{<t}$ and the context $D$, formulated as $p(w_t|w_{<t}, D)$ while IPS splits these two factors. Specifically, proximity value only focuses on the effects of the already generated tokens, i.e., p_value$_t \sim p(w_t|w_{<t})$, and isotropy value pays more attention to the context, i.e., i_value$_t \sim p(w_t|D, (w_{<t}))$ wherein $w_{<t}$ is just used to obtain the undergoing response representation $\mathbf{h}_{RT}$.

## 3 Experiments

**Dataset** We evaluate our method on two commonly-used datasets, DailyDialog (Li et al., 2017) in English and LCCC (Wang et al., 2020) in Chinese. Both of them are open-domain multi-turn dialogue datasets, collected from social media. For LCCC, owing to the academic-level computing resource, we follow previous work (Su et al., 2022), and sample a subset of the dataset, consisting of 100,000 dialogue examples.

**Baselines** Following Wu et al. (2023), we use BART (Lewis et al., 2020) as our backbone. We

evaluate the performance of decoding strategies with different models, including vanilla BART, BART with SimCTG (Su et al., 2022), and BART with SimDRC (Wu et al., 2023). We compare IPS to greedy search, beam search, top-$k$ sampling (Fan et al., 2018), nucleus sampling (Holtzman et al., 2018) and contrastive search (Su et al., 2022).

**Settings** We fine-tune the models on DailyDialog and LCCC datasets for 6k steps and 7k steps, respectively. We use a batch size of 64 and truncate the training samples to a maximum length of 256. The parameters of the models are initialized from HuggingFace libraries and updated by Adam optimizer (Kingma and Ba, 2017) with a learning rate of 3e-5. We adopt the margin values of SimCTG and SimDRC suggested in their work, i.e., $\rho = 0.5$ for SimCTG and $\delta = 0.7, \alpha = 0.3$ for SimDRC. We conduct the isotropic and proximal search with the first $n = 2$ steps adopting top-$k$ sampling ($k = 7$). The weight $\alpha$ is 0.6. We run all experiments with five different seeds and report the average score.

**Evaluation Metrics** Traditional n-gram overlap and text matching metrics such as BLEU (Papineni et al., 2002) and ROUGE (Lin, 2004) are not proper to evaluate plausible output diversity for open-domain dialog systems. Therefore, for automatic evaluation, we choose the following metrics, including BERTScore (Zhang* et al., 2020),

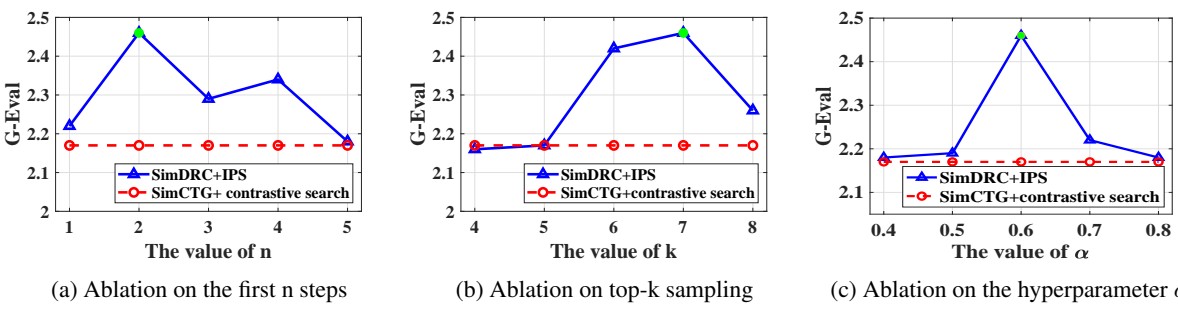

| (a) Ablation on the first n steps | (b) Ablation on top-k sampling | (c) Ablation on the hyperparameter $\alpha$ |

Figure 1: Ablation study on the DailyDialog dataset.

MAUVE (Pillutla et al., 2021), Distinct2/4 (Li et al., 2016a), and G-Eval, an automatic evaluation metric based on GPT3.5 (Liu et al., 2023).

We also conduct a human evaluation with the help of recruited proficient English/Chinese speakers. We randomly sample 100 dialogue examples from DailyDialog and LCCC test sets. For each dialogue context, we generate responses using the aforementioned backbone models (BART, BART+SimCTG, BART+SimDRC) with six different inference strategies. Five annotators are hired independently to measure these samples. Annotators are instructed to give a score ranging from 1 to 5 over the following aspects, including fluency, informativeness, coherence, and semantic coverage[1].

**Results and Discussion** Table 1 lists the automatic evaluation results of the different methods with different decoding strategies. Similar results can be also found in human evaluation, as shown in Table 2. We can see that the models, collaborating with IPS, can produce more semantically consistent(high BERTScores and MAUVE scores) and human-like (high G-Eval scores) responses. Although contrastive search can generate more novel and diverse tokens (high Distinct scores), it usually suffers from the problem of prediction deviation, i.e., the predicted token being weakly related to the main idea of the response. This is also in line with the worse performance of contrastive search on other metrics, such as BERTScore, and G-Eval, indicating that the diverse responses produced by contrastive search are not accurate and human-like enough. Different from contrastive search, IPS tries to concentrate on the core meaning of the response and express it clearly, thus a slightly lower Distinct score is acceptable and expected. Note that IPS still has better distinct scores than other traditional decoding methods since it encourages discrimination and isotropy among utterances.

Although IPS can be directly used with different models and achieve good performance, the models trained with SimDRC are the best testbed for IPS. We can see that SimDRC+IPS can mostly achieve the best performance across the board on both automatic and human evaluation. This is reasonable because the training process of SimDRC is greatly consistent with the search criterion of IPS, and they both push away the inter-utterance features and pull close the intra-utterance features.

**Ablation Study** Figure 1 shows the ablation studies on different components of the method, including the first $n$ steps, the sampling strategy for the first $n$-step decoding, and the weight $\alpha$. As shown in Figure 1(a), our method consistently outperforms the contrastive search no matter the number of first steps. We find some performance drops with the increase of the first-stage sampling steps. We think this is because more generic tokens are selected by traditional search methods, thus weakening the proximity and isotropy of the response. For strategies in the first $n$ steps, we attempt beam search, top-$k$ sampling, and nucleus sampling. We finally select top-$k$ sampling as our first stage's strategy owing to its better performance in the comparisons. Figure 1(b) shows the results of different $k$ values adopted in top-$k$ sampling. We can see that our method exceeds the baseline by a large margin when $k > 5$. The effect of weight $\alpha$ is also studied, as shown in Figure 1(c). Our method consistently outperforms the baseline with the different weights, suggesting the robustness of our method.

**Hyperparameter Analysis** To explore the effects of isotropy and proximity, in our experiments, we introduced a hyperparameter $\beta$ to balance the $p\_value$ and $i\_value$ as:

$$(1 - \beta) \times p\_value - \beta \times i\_value \qquad (6)$$

We tried the effects of $\beta$ ranging from 0.2 to 0.8. We surprisingly found that the balance of proximal

---

[1]Details of human evaluation are in Appendix A.1.

value and isotropy value leads to the best performance, saying $\beta$ equals 0.5. This finding is a bit different from the observations in SimDRC(Wu et al., 2023) which suggests that larger isotropy loss weight is needed to balance the two properties in the training stage. We think this is because our method is a decoding strategy, rather than the training optimization process. The sparse isotropy values would not cause the model bias in the decoding stage. So, the harmonious balance of proximity and isotropy can be simply achieved by giving a moderate value of $\beta$.

## 4 Conclusion

In this work, we present a fine-grained conversational decoding strategy, namely isotropic and proximal search (IPS) to encourage the generation of isotropic and conversational tokens. Superior to existing decoding methods, IPS decouples the previous tokens and the context. Experiments show that our method achieves impressive performance on both automatic and human evaluation.

## Ackonwledgements

This work was supported in part by the InnoHK initiative, the Government of the HKSAR, Laboratory for AI-Powered Financial Technologies.

## Limitations

During the experiments, we found that for a single piece of data in the DailyDialog test set, traditional text decoding methods such as beam search, top-k sampling and beam search take less than 1 second, the contrastive search takes about 5.07s, and the decoding time required by our proposed IPS is about 2.16s. Although our approach takes longer than the traditional text decoding method, our calculation speed is obviously faster than contrastive search. How to further improve the computing speed is still the direction we need to work on.

## Ethics Statement

In this work, we use publicly released datasets to auxiliary our dialogue response generation. Generally, these previous works have considered ethical issues when creating the datasets. We have manually checked some samples for the datasets we used in this work, and do not find any obvious ethical concerns, such as violent or offensive content. We will also release the source decoding code

with friendly instructions to support its correct use. However, we still need to emphasize that text generation is not as controllable as we think. It still would generate some novel or unexpected words occasionally. We may take actions to decrease generation diversity to alleviate this problem.

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

# A  Appendix

## A.1  Human Evaluation Instructions

Please rate the quality of the generated response based on the given dialogue context and the target response over the following aspects: (1) Fluency; (2) Informativeness; (3) Coherence; (4) Semantic Coverage. We provide some instructions for your rating.

### A.1.1  Fluency

This measures whether the generated text has no formatting problems, capitalization errors, or obviously ungrammatical sentences (e.g., fragments, missing components) that make the text difficult to read. The definitions of different scores are:

- **5**: The text is fluent, grammatically correct, and has no errors. It is easy to read.

- **4**: The text is grammatically correct but has a few spelling or capitalization errors, which does not affect your understanding.

- **3**: The text has minor errors in both grammar and spelling. The errors slightly affect your understanding.

- **2**: The text has major errors in both grammar and spelling. The errors make the text hard to read.

- **1**: The text does not make sense and it is unreadable.

### A.1.2 Informativeness

This measures whether the generated text has diverse, informative, novel, or logically related content. The definitions of different scores are:

- **5**: The text contains very diverse, informative, and novel content. It is enjoyable to read the text.

- **4**: The text contains many informative and novel contents. (Choose this score when you hesitate between 3 and 5.)

- **3**: The text contains some new information but also contains a few repetitions of the context.

- **2**: The text only contains a few informative and new terms. (Choose this score when you hesitate between 1 and 3.)

- **1**: The text is dull, repetitive, and has no new information. All contents are from the dialogue context.

### A.1.3 Coherence

This measures whether the generated text is semantically and factually consistent with the dialogue context. The definitions of different scores are:

- **5**: The text is semantically, factually, and topically consistent with the dialogue context. All contents of the text are related to the source text or can be inferred from the source.

- **4**: The text is very related to the context but has minor inconsistencies or contradictions that do not affect its overall relevance.

- **3**: The text is related to the context but has some obvious inconsistencies and contradictions.

- **2**: The text is slightly consistent with the context. Many inconsistencies and contradictions in the context can be found.

- **1**: The text is totally inconsistent with the context. It semantically or factually contradicted the context.

### A.1.4 Semantic Coverage

This measures how many semantic content units from the target response are covered by the generated text. The definitions of different scores are:

- **5**: All semantic content units of the target text can be found in the generated text. They are semantically consistent.

- **4**: Most of the content units of the target text can be found from the generated text while a few missing units do not affect the overall coverage.

- **3**: Some semantic content units can be found in the generated text but also miss some important units.

- **2**: Most of the semantic content units are not covered. Only a few insignificant units can be found in the generated text.

- **1**: The text does not have any overlapping semantic content units with the target text.

We recruit five human workers to annotate 3,600 samples. To make sure the workers are fairly paid, we pay 0.1 dollars for each sample. Therefore, the total amount spent on participant compensation is 360 dollars. The annotators take 24 hours to finish the task, suggesting the hourly wage for each worker is 15 dollars.

## A.2 More Details of the Task

### A.2.1 Evaluation of G-EVAL Score

The API we used to test G-EVAl is *gpt-3.5-turbo*, and the following is the prompt (Liu et al., 2023):

---

You will be given a conversation between two individuals. You will then be given one potential response for the next turn in the conversation. Your task is to give a final score for utterance. Please make sure you read and understand these instructions carefully.

The evaluation aspects are:

1. Engagingness: Is the response dull or interesting?

2. Naturalness: This measures whether the generated text has no formatting problems, capitalization errors, or obviously ungrammatical sentences to read.

3. Informativeness: This measures whether the generated text has diverse, informative, novel, or logically related content.

4. Coherence: This measures whether the generated text is semantically and factually consistent with the dialogue context.

The evaluation steps are:

1. Read the conversation, the corresponding label, and the response carefully.

2. Considering the above evaluation aspects, return a comprehensive final score ranging from 1 to 5 for each conversation.

3. Please only return 1 overall score, without any extra text descriptions. The return format should be like *Score:1*.

Now please read the following conversation, and return the score.

---

### A.2.2 More Experimental Results

Table 2 lists the results of human evaluation.

### A.3 Surface-level Analysis

### A.3.1 Score Distribution According to the Length of the Previous Context

Table 3 and Table 4 illustrate the relations between the context length and the human evaluation metrics while using the IPS (the above one) and beam search (the below one) decoding strategies. Observing the table, when the context length is particularly short (<10), we speculate that the context may consist of simple greetings or introductions, resulting in lower difficulty of generation and thus higher scores. When the context length varies in the range of approximately 10 to 40, due to differences in the complexity of context content and semantics, the scores exhibit a fluctuating trend. As the length continues to increase, the information provided by the previous context becomes richer, leading to improved effectiveness of both decoding methods. We also note that when faced with exceptionally long contexts, the generation quality of IPS is superior to the baselines.

### A.3.2 Utterance Length Analysis

Table 5 shows that both IPS and contrastive search tend to produce shorter sentences than traditional methods. We explain in the main text that by incorporating isotropy, achieved through contrastive

search and IPS, redundancy is minimized, resulting in more concise generated text compared to previous methods. Considering the nature of the conversation, our IPS strategy expects proximity and does not enlarge the token distance in the same utterance, thus responses of IPS are slightly longer than that of contrastive search.

### A.4 Qualitative Analysis

### A.4.1 Instances Illustration

Some examples are presented to illustrate the effect of our IPS search.

In summation, according to Table 6 and Table 7, some qualitative observations are as follows:

- Replies generated by IPS are more natural and accurate.

- IPS tends to generate relatively concise responses.

- With more complex previous contexts, we observed that IPS does not prioritize shortening the length of response. IPS can generate responses that are more in line with the situation based on the characteristics of the conversation.

### A.5 Cosine Similarity Heatmap

To ensure utterances generated by our IPS are isotropic and proximal, and observe the representations produced by different decoding methods, we showcase the cosine similarity matrix of token representations correspondingly.

The larger color difference between different sentences represents greater isotropy, indicating discrimination among utterances; while the darker the color within the same sentence, the greater the proximity, conveying a more concentrated thought.

Choosing SimDRC as the backbone model, cosine similarity heatmaps of different inference methods are shown as follows. Tokens generated by IPS exhibit brighter colors in the heatmap, indicating increased proximity within the same sentence, while tokens from IPS showcase darker colors for different sentences, signifying greater isotropy. Contrastingly, traditional methods like beam search showed anisotropy(i.e. features occupy a narrow cone in the vector space, thus leading to the problem of degeneration.) in the figures.

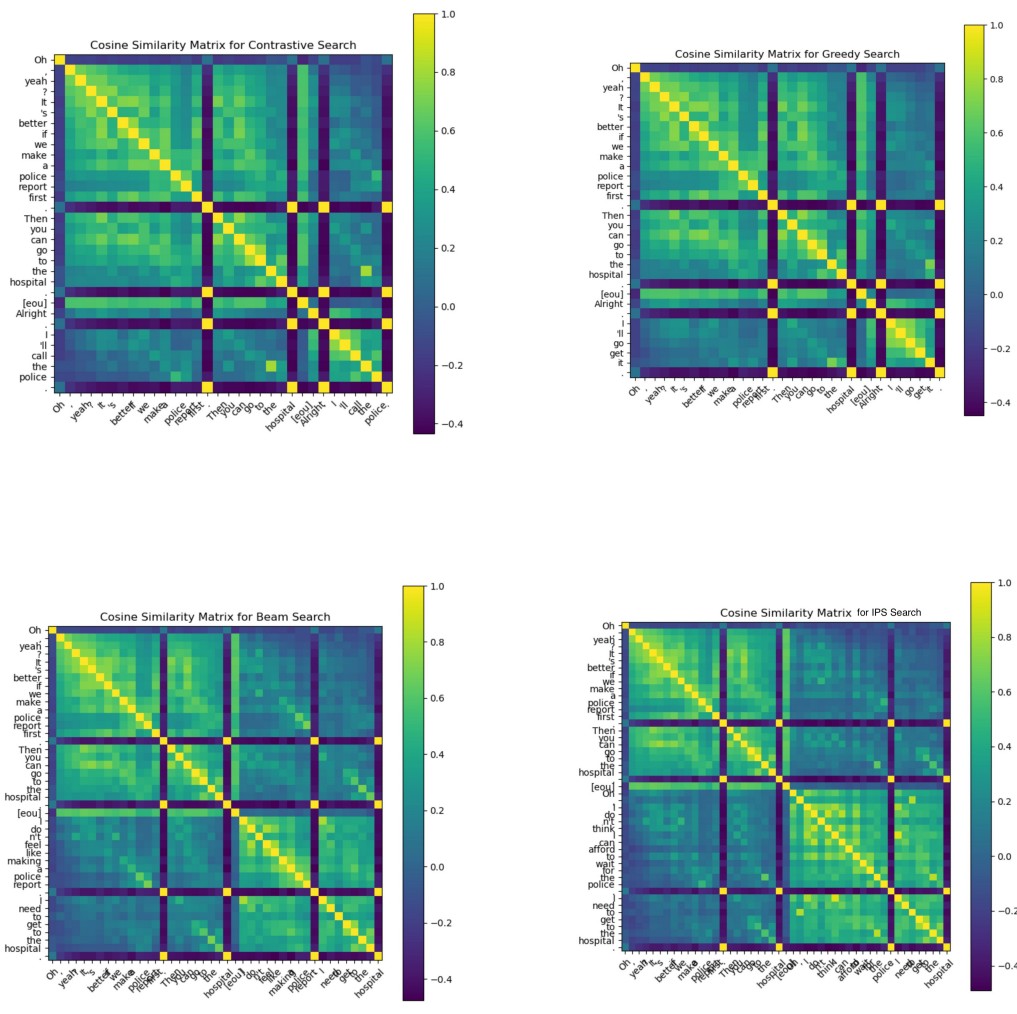

Figure 2: An image of a cosine similarity heatmap

| Model | Strategy | DailyDialog | | | | LCCC | | | |
|---|---|---|---|---|---|---|---|---|---|
| | | Fluency | Info. | Coherence | SC | Fluency | Info. | Coherence | SC |
| BART | greedy | 4.636 | 3.302 | 3.362 | 2.386 | 4.626 | 2.810 | 3.414 | 1.996 |
| | beam | 4.584 | 3.362 | 3.508 | 2.390 | 4.452 | 2.950 | 3.278 | 2.054 |
| | top-$k$ | 4.634 | 3.416 | 3.554 | 2.432 | 4.554 | 2.954 | 3.384 | 2.072 |
| | nucleus | 4.666 | 3.478 | 3.578 | 2.420 | 4.602 | 3.042 | 3.434 | 3.358 |
| | contrastive | 4.678 | 3.406 | 3.476 | 2.416 | 4.560 | 2.888 | 3.358 | 2.118 |
| | IPS | **4.710** | **3.562** | **3.768** | **2.566** | **4.718** | **3.152** | **3.622** | **2.184** |
| SimCTG ($\rho = 0.5$) | greedy | 4.652 | 3.288 | 3.362 | 2.394 | 4.622 | 2.884 | 3.580 | 2.026 |
| | beam | 4.696 | 3.404 | 3.446 | 2.390 | 4.602 | 2.918 | 3.224 | 2.040 |
| | top-$k$ | 4.718 | 3.398 | 3.406 | 2.414 | 4.554 | 2.970 | 3.464 | 2.040 |
| | nucleus | 4.734 | 3.372 | 3.348 | 2.386 | 4.582 | 2.938 | 3.548 | 2.068 |
| | contrastive | 4.712 | 3.304 | 3.31 | 2.332 | 4.582 | 2.882 | 3.456 | 2.076 |
| | IPS | **4.758** | **3.586** | **3.578** | **2.584** | **4.708** | **3.084** | **3.688** | **2.176** |
| SimDRC ($\delta = 0.7$, $\alpha = 0.3$) | greedy | 4.774 | 3.632 | 3.484 | 2.684 | 4.634 | 2.920 | 3.390 | 1.990 |
| | beam | 4.820 | 3.580 | 3.404 | 2.586 | 4.620 | 2.920 | 3.632 | 2.034 |
| | top-$k$ | 4.854 | 3.614 | 3.396 | 2.618 | 4.590 | 2.950 | 3.572 | 2.048 |
| | nucleus | 4.864 | 3.622 | 3.450 | 2.600 | 4.588 | 2.930 | 3.628 | 2.088 |
| | contrastive | 4.872 | 3.692 | 3.798 | 2.694 | 4.594 | 2.890 | 3.582 | 1.984 |
| | IPS | **4.892** | **3.768** | **3.942** | **2.826** | **4.712** | **3.120** | **3.734** | **2.332** |

Table 2: Results of human evaluation on DailyDialog and LCCC datasets, where SC means the semantic coverage, info. means informativeness.

| Length | Fluency | Informativeness | Coherence | Semantic Coverage | num |
|---|---|---|---|---|---|
| [0,10) | 4.94 | 4.56 | 4.67 | 3.06 | 9 |
| [10,20) | 4.93 | 3.5 | 3.62 | 2.77 | 13 |
| [20,30) | 4.73 | 4 | 4 | 3.09 | 11 |
| [30,40) | 4.75 | 3.56 | 4 | 2.67 | 12 |
| [40,50) | 4.85 | 3.67 | 3.56 | 2.28 | 9 |
| [50,75) | 4.95 | 3.52 | 3.61 | 2.45 | 17 |
| [75,100) | 4.8 | 3.52 | 3.79 | 2.84 | 15 |
| over 100 | 4.93 | 3.79 | 4.21 | 2.96 | 14 |

Table 3: Relations between the context length and the human evaluation metrics while using the IPS.

## A.6 Examples of Generated Texts

For non-native Chinese speakers, translations of Table 9 are presented in Table 10. The quality of the LCCC dataset still requires optimization, as it contains numerous colloquial and slang expressions. We are not professional translators, and in our attempts, we noticed that the translated meanings sometimes diverged from the original Chinese. We apologize for the inconvenience.

| Length | Fluency | Infomrativeness | Coherence | Semantic Coverage | num |
|--------|---------|-----------------|-----------|-------------------|-----|
| [0,10) | 4.89 | 4.44 | 4.33 | 2.78 | 9 |
| [10,20) | 4.77 | 3.31 | 3.15 | 2.46 | 13 |
| [20,30) | 4.55 | 3.86 | 3.41 | 3 | 11 |
| [30,40) | 4.88 | 3.29 | 3.13 | 2.42 | 12 |
| [40,50) | 4.88 | 3.11 | 2.89 | 1.83 | 9 |
| [50,75) | 4.82 | 3.43 | 3.17 | 2.43 | 17 |
| [75,100) | 4.78 | 3.48 | 3.45 | 2.5 | 15 |
| over 100 | 4.93 | 3.5 | 3.64 | 2.43 | 14 |

Table 4: Relations between the context length and the human evaluation metrics while using the beam search.

| Sentence_Len | dd-drc | dd-ctg | dd-bart | lccc-drc | lccc-ctg | lccc-bart |
|--------------|--------|--------|---------|----------|----------|-----------|
| Greedy | 11.33 | 11.39 | 10.98 | 14.05 | 13.76 | 14.60 |
| Beam | 12.56 | 12.65 | 12.24 | 11.89 | 11.69 | 11.88 |
| Top_k | 13.10 | 13.21 | 12.71 | 12.39 | 12.12 | 12.40 |
| Top_p | 12.76 | 13.60 | 13.17 | 12.55 | 12.33 | 12.56 |
| Simctg | 9.02 | 9.05 | 8.72 | 9.50 | 9.17 | 8.88 |
| IPS | 9.59 | 10.31 | 9.13 | 9.95 | 9.46 | 9.14 |

Table 5: Sentence-level Analysis

| | |
|--|--|
| context | A: I have a cell phone in my car. Now it's probably on the floor on the passenger side. Why don't you get it for me, and then I can call the police? B: Alright A: It doesn't work. It looks like it's broken. I need to get to a hospital. You should drive me there. B: Oh, yeah? It's better if we make a police report first. Then you can go to the hospital. |
| label | "Damn it! I'm injured here. We could wait all day for the police." |
| greedy | "Alright. I'll go get it." |
| beam/top-p/top-k | "I don't feel like making a police report. I need to get to the hospital." |
| contrastive | "Alright. I'll call the police." |
| IPS | "Oh, I don't think I can afford to wait for the police. I need to get to the hospital." |

Table 6: Qualitative Instance A

| | |
|--|--|
| context | A: Good afternoon! Did you call for a plumber? B: Yes, yes I did. Please come in! I'm so glad you came! This old house is falling apart! Come on into the bathroom. See, here, there's water leaking everywhere! |
| greedy | "Well, it's leaking because of the leaking faucet in the shower." |
| beam/top-p | "Well, it's leaking because of the leaking faucet in the shower." |
| top-k | "I'm sorry, madam, but I don't think that water is leaking. I'm going to have to check it out." |
| contrastive | "Well, it's leaking all over! I don't know what the plumber can do about it!" |
| IPS | "Well, it's leaking because of the leaking faucet in the shower. I guess the water is backed up." |

Table 7: Qualitative Instance B

| | |
|---|---|
| content | A: I didn't receive an unemployment check this week and need to know why.
B: Did you mail us your last Continued Claim Form?
A: I am afraid that I forgot to send it in.
B: We cannot pay you unless you turn that form in every two weeks.
A: Is it too late to mail the Continued Claim Form in now? |
| greedy | I will mail it in right now. |
| beam search | It's too late to mail it in now. You will receive your check this week. |
| top-k sampling | I will mail it in right now if you can't wait. |
| nucleus search | It's too late to mail it in now. You will receive your check this week. |
| contrastive search | Yes, you can do it now. |
| IPS | Yes, it is. |
| content | A: Hello, this is Mike, Kara.
B: Mike! Good to hear from you. How are you?
A: Everything is fine, and how are you?
B: Things are going well with me.
A: Kara, I had fun the other night at the movies and was wondering if you would like to go out again this Friday. |
| greedy | I was thinking of it, but I am not feeling too well. I have to go to the hospital for a couple of days. |
| beam search | That sounds great. I would love to go out with you |
| top-k sampling | I would love to, but I have to work late on Friday. |
| nucleus search | That sounds great. I would love to go out with you. |
| contrastive search | Mike, I was thinking of it. |
| IPS | That sounds great. I would love to. |
| content | A: What do you think I should do, doctor? |
| greedy | I think you should have a good rest for three days. |
| beam search | I think you'd better have a good rest for three days |
| top-k sampling | I think you should go to see the head of the family. |
| nucleus search | I think you'd better have a good rest for three days. |
| contrastive search | Well, it's hard to say. You look a little pale, don't you? |
| IPS | Well, you should have a good rest for three days. Then you'll feel better. |

Table 8: Results of generation texts of DailyDialog.

| | |
|---|---|
| content | A: 姐做的好喝
B: 这句姐喜欢
A: 如果做给俺喝，那绝对是美味啊，人间极品，佳肴中滴佳肴
B:家里有蕃茄但现在姐不想做哩 |
| greedy | 我也喜欢喝，但是我不会做给你喝 |
| beam search | 我也不想做，我喜欢吃番茄 |
| top-k sampling | 我也喜欢吃这种美味 |
| nucleus search | 我也不想做了，就是喜欢吃" |
| contrastive search | 我也不想做，就是喜欢吃这种东西而已 |
| IPS | 我也不想做，但是我喜欢吃 |
| content | A: 魔都都不冷了？
B: 这两天有点反常吧…估计过两天会冷下去… |
| greedy | 我们这里也是 |
| beam search | 我们这里也是这样的天气 |
| top-k sampling | 我们这里也是这样的天气 |
| nucleus search | 这两天有点冷，但是也不是特别冷 |
| contrastive search | 这两天好冷啊，估计明天开始就冷 |
| IPS | 这两天天气不错，不过今天有点冷 |
| content | A: 我饿了。 |
| greedy | 我也饿了。 |
| beam search | 我也想吃，但是没有时间。 |
| top-k sampling | 我饿了，我要去吃饭了。 |
| nucleus search | 你怎么这么早就睡了。 |
| contrastive search | 我也饿了。 |
| IPS | 我也饿了，你吃了吗? |

Table 9: Results of generation texts of LCCC.

| | |
|---|---|
| content | A: You are good at beverage |
| | B: I like your comment. |
| | A: If made for me to drink, it would definitely be delicious, an exquisite delicacy among delicacies. |
| | B: There are tomatoes at home, but right now, I just don't feel like dealing with them. |
| greedy | I also like to drink, but I won't make it for you to drink. |
| beam search | I don't want to cook either, I enjoy eating tomatoes. |
| top-k sampling | I also enjoy eating this delicious treat. |
| nucleus search | I don't want to make it anymore, I just like to eat it |
| contrastive search | I don't want to cook either, I simply enjoy eating this kind of thing. |
| IPS | I don't want to do it either, but I'd love to eat. |
| content | A: Is Shanghai not cold anymore? |
| | B: These two days have been a bit abnormal... I estimate it will get colder in the next two days |
| greedy | We here the same |
| beam search | We have the same weather here |
| top-k sampling | We have the same weather here |
| nucleus search | It's been a bit cold these two days, but it's not particularly cold either |
| contrastive search | It's been so cold these two days. It's expected to start cold tomorrow |
| IPS | The weather has been good these two days, but it's a bit cold today |
| content | A: I'm hungry |
| greedy | I'm also hungry |
| beam search | I want to eat too, but I don't have time |
| top-k sampling | I'm hungry, I'm going to eat now |
| nucleus search | Why did you go to bed so early |
| contrastive search | I'm also hungry |
| IPS | I'm also hungry, have you eaten? |

Table 10: Translation of generation texts of LCCC.