# OpenReview forum: "Fine-grained Conversational Decoding via Isotropic and Proximal Search"
_EMNLP/2023/Conference — EMNLP 2023 Main_

### Official Review · Reviewer_T3Fd · 2023-07-30

**Soundness:** 4

**Ethical Concerns:**

Yes

**Excitement:**

4: Strong: This paper deepens the understanding of some phenomenon or lowers the barriers to an existing research direction.

**Paper Topic And Main Contributions:**

This paper introduces a new decoding method that is argued to be more suitable for dialogue response generation. The authors based off their method on the finding that a good dialogue feature space should follow the rules of locality and isotropy. To achieve this, the authors designed a decoding objective that optimizes for locality (by maximizing the average cosine similarity between the representation of newly generated word and the previously generated words) and isotropy (by minimizing the cosine similarity between the representation of words generated sofar and that of the past utterances). The paper then used two datasets to compare their method against other decoding methods (e.g. contrastive search), and show that it outperforms others under both automatic evaluations (BERTScore, MAUVE, G-EVAL) and human evaluation.

**Questions For The Authors:**

Question A: How does/can IPS avoid degradation of the generation, if generating repetitive rephrases seems to be favored by $p_\text{value}$ and not disencouraged by $i_\text{value}$?

Question B: Can you provide more direct/concrete way of showing how IPS generated responses are more proximal and isotropic? Most experiments shown in this paper are too "end-to-end".

**Reasons To Accept:**

The paper proposes a new decoding method for dialogue responses. This can be used in parallel with other methods (e.g. better modeling, better prompts) and could be helpful in building better dialogue systems in the future. The strength of this paper includes:

1. This paper defines a new decoding algorithm that promotes proximity and isotropy, which are found to be important for conversational responses.
2. Both automatic evaluations (2 datasets with 5 competitive baselines) and human evaluations show that IPS generated responses are better.

**Reasons To Reject:**

1. The setup/purpose of the ablation study is confusing. Why is Figure 1a,b comparing G-EVAL but Figure 1c comparing MAUVE? Why are we even looking at G-EVAL/MAUVE as opposed to other metrics? Why are we comparing SimDRC+IPS against *SIMCTG*+constrastive search, but not SimDRC+IPS against SimDRC+contrastive search?
2. Lack of direct analysis of how IPS improves proximity and isotropy. The automatic metrics used (in both main experiment and ablation), such as "BERTScore, MAUVE, G-EVAL" are all very generic. While IPS does show improvement under these metrics, it is unclear if it is due to the utterances being more proximal and isotropic, or other reasons.
3. Only five annotators are involved in human evaluation, which may be too few as no statistical significance is measured.

**Reproducibility:**

4: Could mostly reproduce the results, but there may be some variation because of sample variance or minor variations in their interpretation of the protocol or method.

**Reviewer Confidence:**

3: Pretty sure, but there's a chance I missed something. Although I have a good feel for this area in general, I did not carefully check the paper's details, e.g., the math, experimental design, or novelty.

---

> ### Author Rebuttal · Authors · 2023-08-28
>
> Thank you for taking the time to review our paper and for providing valuable feedback. We appreciate your thoughtful comments and suggestions, which help us to improve the quality of our work.
>
> ---
>
> **Concern:** Settings in the ablation study
>
> **Response:**  Regarding your concern in the ablation study, we establish G-EVAL as a dependable metric due to its strong correlation with human judgments, attributed to its LLM-based nature. Additionally, our method demonstrates superiority in the G-EVAL metric during the ablation study on the alpha hyperparameter as well. We chose MAUVE here since the performance gap on this metric is larger and more intuitive. Given the restriction on sharing image links, we're currently presenting the results in a tabular format. The baseline G-Eval score is 2.17 as shown in Table 1.
>
> |alpha | G-EVAL |
> |:-----:|:------:|
> |0.4| 2.18   |
> |0.5| 2.19   |
> |0.6| 2.46   |
> |0.7| 2.22   |
> |0.8| 2.18   |
>
> We conducted ablation experiments of different hyperparameters on MAUVE and other metrics as well. All results consistently indicate that our IPS is the superior one.
>
> ---
>
> **Concern:** Why are we comparing *SimDRC+IPS* against  *SIMCTG+constrastive search* ?
>
> **Response:**  We appreciate the suggestion about the baseline selection. Note that the contrastive search is designed for SimCTG and IPS is inspired by SimDRC. In this context, we compared *SimDRC+IPS* against *SimCTG+contrastive search* by the intuition that the decoding strategies should work better with the corresponding models.
>
> For a comprehensive evaluation, we also conducted the comparisons to *SimDRC+nucleus*, *SimDRC+contrastive search* and *SimCTG+IPS*. All comparison results show that *SimDRC+IPS* is the better one.
>
> ---
>
> **Concern:** how IPS improves proximity and isotropy & Question B
>
> **Response:** In our research, the proposed decoding strategy doesn't alter the model's feature distribution; instead, it focuses on identifying tokens within the distribution, meeting specific requirements based on proximity and isotropy properties post-fine-tuning. This distinction underscores why our decoding strategy combined with SimDRC yields the best results. To address your question of whether IPS generates more isotropic and proximal tokens compared to other methods, we conducted a comparison by examining the cosine similarity matrices of tokens generated by different decoding strategies with the same context. As we cannot insert images here, we depict the outcomes: tokens generated by IPS exhibit brighter colors in the image, indicating increased proximity within the same sentence, while tokens from IPS showcase darker colors for different sentences, signifying greater isotropy. Contrastingly, traditional methods like beam search showed anisotropy in the images. You can refer to Figure 4 in "Learning Locality and Isotropy in Dialogue Modeling (SimDRC)".
>
> We appreciate your inquiry, and in the final version, we intend to include cosine similarity heatmaps, offering readers a more intuitive experience of IPS.
>
> ---
>
>
> **Concern:** The ablation study of isotropy and proximity & Question A
>
> **Response:** We sincerely appreciate your suggestions about the ablation study. In our experiments, we actually introduced a hyperparameter to balance the p_value and i_ value as $(1-\beta) \times p\\_value - \beta \times i\\_value$. We explored the effects of $\beta$ ranging from 0.2 to 0.8. We surprisingly found that the balance of proximal value and isotropy value leads to the best performance, saying $\beta$ equals 0.5. This finding is a bit different from the observations in SimDRC which suggests that larger isotropy loss weight is needed to balance the two properties in the training stage. We think this is because our method is a decoding strategy, rather than the training optimization process. The sparse isotropy values would not cause the model bias in the decoding stage. So, the harmonious balance of proximity and isotropy can be simply achieved by giving a moderate value of $\beta$. We will add the discussions about how to balance the values of proximity and isotropy in the revision.
>
> ---
>
> **Concern:** amount of annotators
>
> **Response:** Regarding the issue of the number of annotators: Generally, five annotators are enough. For instance, five annotators are hired in both "A Contrastive Framework for Neural Text Generation (SimCTG)" and "Learning Locality and Isotropy in Dialogue Modeling (SimDRC)" papers.
>
> ---
>
> We sincerely thank you again for your suggestions!

---

### Official Review · Reviewer_nWpT · 2023-08-04

**Typos Grammar Style And Presentation Improvements:** "Coherenc" -> "Coherence" in Table 2 …
**Soundness:** 4

**Excitement:**

4: Strong: This paper deepens the understanding of some phenomenon or lowers the barriers to an existing research direction.

**Missing References:**

N/A

**Paper Topic And Main Contributions:**

This paper is motivated from the locality and isotropy for modeling dialogue, and proposes isotropic and proximal search (IPS). Specifically, there are additional terms on decoding process, which are proximal value (avg. distance between candidate token and the already generated tokens) and isotripic value (avg. similarity between undergoing response and all utterances). The authors evaluated on various metric including human evaluation and the proposed decoding method shows prominent performance.

**Questions For The Authors:**

- Question A: Do the authors analyze the score distribution according to the length of previous context? In Table 3, I think the generated samples of third content looks quite similar, thus I wonder if the proposed decoding strategy works better on long previous context.
- Question B: Could the authors write English-translated version together in Table 4?

**Reasons To Accept:**

This work nicely borrows the locality and isotropy concept and adopts them into decoding process. It is sufficiently intuitive and detailed experimental settings and ablations are evidential. Furthermore, although the IPS is slower than traditional methods (e.g., beam search, top-k sampling), as the authors mentioned in Limitation section, it is still faster than contrastive search. This work would enlighten future possibilities of conversational modeling.

**Reasons To Reject:**

Ablation studies exploring the effect of respective of proximal and isotropic values need to be conducted. Others are reasonable to understand, so please refer to the questions in the next section.

**Reproducibility:**

5: Could easily reproduce the results.

**Reviewer Confidence:**

3: Pretty sure, but there's a chance I missed something. Although I have a good feel for this area in general, I did not carefully check the paper's details, e.g., the math, experimental design, or novelty.

---

> ### Author Rebuttal · Authors · 2023-08-28
>
> Thank you for taking the time to review our paper and for providing valuable feedback. We appreciate your thoughtful comments and suggestions, which help us to improve the quality of our work.
>
> ---
>
> **Concern:** The ablation study of isotropy and proximity
>
> **Response:** We sincerely appreciate your suggestions about the ablation study. In our experiments, we actually introduced a hyperparameter to balance the p_value and i_ value as $(1-\beta) \times p\\_value - \beta \times i\\_value$. We explored the effects of $\beta$ ranging from 0.2 to 0.8. We surprisingly found that the balance of proximal value and isotropy value leads to the best performance, saying $\beta$ equals 0.5. This finding is a bit different from the observations in SimDRC which suggests that larger isotropy loss weight is needed to balance the two properties in the training stage. We think this is because our method is a decoding strategy, rather than the training optimization process. The sparse isotropy values would not cause the model bias in the decoding stage. So, the harmonious balance of proximity and isotropy can be simply achieved by giving a moderate value of $\beta$. We will add the discussions about how to balance the values of proximity and isotropy in the revision.
>
> ---
>
> **Concern:** Score distribution according to the length of the previous context
>
> **Response:** Regarding your concern about score distribution according to the length of the previous context, Our statistics are as follows：
>
> | Length   | Fluency | Informativeness | Coherence | Semantic Coverage | num  |
> |----------|---------|-----------------|-----------|-------------------|------|
> | [0,10)   | 4.94    | 4.56            | 4.67      | 3.06              | 9    |
> | [10,20)  | 4.93    | 3.5             | 3.62      | 2.77              | 13   |
> | [20,30)  | 4.73    | 4               | 4         | 3.09              | 11   |
> | [30,40)  | 4.75    | 3.56            | 4         | 2.67              | 12   |
> | [40,50)  | 4.85    | 3.67            | 3.56      | 2.28              | 9    |
> | [50,75)  | 4.95    | 3.52            | 3.61      | 2.45              | 17   |
> | [75,100) | 4.8     | 3.52            | 3.79      | 2.84              | 15   |
> | over 100 | 4.93    | 3.79            | 4.21      | 2.96              | 14   |
>
> | Length   | Fluency | Infomrativeness | Coherence | Semantic Coverage | num  |
> |----------|---------|-----------------|-----------|-------------------|------|
> | [0,10)   | 4.89    | 4.44            | 4.33      | 2.78              | 9    |
> | [10,20)  | 4.77    | 3.31            | 3.15      | 2.46              | 13   |
> | [20,30)  | 4.55    | 3.86            | 3.41      | 3                 | 11   |
> | [30,40)  | 4.88    | 3.29            | 3.13      | 2.42              | 12   |
> | [40,50)  | 4.88    | 3.11            | 2.89      | 1.83              | 9    |
> | [50,75)  | 4.82    | 3.43            | 3.17      | 2.43              | 17   |
> | [75,100) | 4.78    | 3.48            | 3.45      | 2.5               | 15   |
> | over 100    | 4.93    | 3.5             | 3.64      | 2.43              | 14   |
>
> These two tables illustrate the relations between the context length and the human evaluation metrics while using the IPS (the above one) and beam search (the below one) decoding strategies. Observing the table, when the context length is particularly short (<10), we speculate that the context may consist of simple greetings or introductions, resulting in lower difficulty of generation and thus higher scores. When the context length varies in the range of approximately 10 to 40, due to differences in the complexity of context content and semantics, the scores exhibit a fluctuating trend. As the length continues to increase, the information provided by the previous context becomes richer, leading to improved effectiveness of both decoding methods. We also note that when faced with exceptionally long contexts, the generation quality of IPS is superior to the baselines.
>
> ---
>
> **Concern:** problems in presentation
>
> **Response:** Thank you for pointing out the typo and the missing of English translation in Table 4. We will revise the problems in the revision.
>
> ---
>
> Thank you again for your valuable suggestions!

---

### Official Review · Reviewer_ybjD · 2023-08-04

**Soundness:** 4

**Excitement:**

4: Strong: This paper deepens the understanding of some phenomenon or lowers the barriers to an existing research direction.

**Paper Topic And Main Contributions:**

The paper presents a conversational decoding strategy named isotropic and proximal search (IPS). As the name itself suggests, the decoding strategy is based on the concepts of locality and isotropy, two key features that are shown to be essential for generating high-quality dialogue responses. IPS has a parameter alpha that controls for the weight assigned to these two components; when alpha=1, the IPS behaves as greedy decoding. The authors assess the performance of IPS on two datasets (English and Chinese) and different variants of BART and decoding strategies.

**Questions For The Authors:**

- Question A: Did the authors find any surface-level pattern in the utterances generated by IPS (utterance length, unique tokens, etc.)?
- Question B: Do the authors have any intuition about why human annotators disagree way more when evaluating the informativeness of generated utterances in the DailyDialog task compared to LCCC (0.56 vs. 0.78)? All the other metrics instead (fluency, coherence, semantic coverage) have similar values in the two settings.
- Question C: Did the authors inspect more examples compared to the ones reported in Table 3? Did they get any qualitative insight into the main advantages of IPS-generated utterances?

**Reasons To Accept:**

The paper is well-written and easy to follow. It presents an original conversational decoding strategy and the authors evaluate its effectiveness against several other decoding strategies. The experimental setup is clear and well-described. The authors also conducted a human evaluation on a subset of the generated utterances and perform interesting ablation studies. Overall, the results seem to suggest that IPS outperforms other decoding strategies in most of the settings and the evaluation metrics employed.

**Reasons To Reject:**

Although the paper has the potential to represent an original and significant contribution to the field, a few issues deserve a closer look. The differences reported in Tables 1 and 2 are sometimes minor, and statistical significance tests are needed to validate the claim that IPS outperforms other decoding strategies. In case it turns out that some differences are not significant, IPS still represents a novel and original contribution; however, it is important to know which differences are significant to adjust the overall claim of the paper.
In the conclusions, the authors mention: “Experiments show that our method achieves impressive performance on both automatic and human evaluation”, and in the abstract they say: “Experiments show that our approach significantly outperforms existing decoding strategies”. These are clearly too strong claims that have to be adjusted depending on the results of significance tests.
More qualitative examples and surface-level statistics (utterance length, vocabulary coverage, etc.) would be helpful to assess the effectiveness of IPS. The examples reported in Table 3 do not show a clear qualitative advantage of IPS over other decoding strategies, while Table 1 and Table 2 illustrate that IPS (almost) always outperforms other methods against both automatic and human-based metrics. The examples reported in Table 3 suggest that it is advisable to have a closer look at possible limitations in the evaluation metrics used and/or biases during the human annotation procedure.

**Reproducibility:**

4: Could mostly reproduce the results, but there may be some variation because of sample variance or minor variations in their interpretation of the protocol or method.

**Reviewer Confidence:**

3: Pretty sure, but there's a chance I missed something. Although I have a good feel for this area in general, I did not carefully check the paper's details, e.g., the math, experimental design, or novelty.

**Typos Grammar Style And Presentation Improvements:**

The title sounds a bit too vague, maybe it is worth mentioning the concepts of isotropic and proximal search there.

---

> ### Author Rebuttal · Authors · 2023-08-28
>
> Thank you for taking the time to review our paper and for providing valuable feedback.  We appreciate your thoughtful comments and suggestions, which help us to improve the quality of our work.
>
> ---
>
> **Concern:** significance tests and statement issue
>
> **Response:** As our experimental settings, we ran all experiments with three different random seeds and reported the average scores. We also conducted the significance tests on the metrics of BertScore, MAUVE, and G-Eval. The results show that our contribution is significant with the sign test p-value < 0.05. We will also carefully consider your comments regarding the issue of 'outperform' and will revise the writing in the revision to avoid overstatement.
>
> ---
>
> **Concern:**  surface-level pattern in the utterances
>
> **Response:** We appreciate your suggestions related to statistics and analysis of surface-level indicators. We conducted experiments  and the results are shown in the following table:
>
> | Sentence_Len | dd-drc | dd-ctg | dd-bart | lccc-drc | lccc-ctg | lccc-bart  |
> |:------------:|:------:|:------:|:-------:|:--------:|:--------:|:----------:|
> | Greedy       | 11.33 | 11.39  | 10.98   | 14.05 | 13.76    | 14.60      |
> | Beam         | 12.56  | 12.65| 12.24   | 11.89    | 11.69| 11.88      |
> | Top_k        | 13.10| 13.21  | 12.71   | 12.39| 12.12    | 12.40      |
> | Top_p        | 12.76| 13.60  | 13.17   | 12.55| 12.33    | 12.56      |
> | Simctg          | 9.02   | 9.05| 8.72    | 9.50     | 9.17| 8.88       |
> | IPS          | 9.59   | 10.31| 9.13    | 9.95     | 9.46| 9.14       |
>
> The results show that both IPS and contrastive search tend to produce shorter sentences than traditional methods. We explain in the main text that by incorporating isotropy, achieved through contrastive search and IPS, redundancy is minimized, resulting in more concise generated text compared to previous methods. Considering the nature of the conversation, our IPS strategy expects proximity and doesn’t enlarge the token distance in the same utterance, thus responses of IPS are slightly longer than that of contrastive search.
>
> ---
>
> **Concern:** Qualitative advantages of IPS over other decoding strategies
>
> **Response:** Some qualitative observations are as follows:
>
> 1. Replies generated by IPS are more natural and accurate.
> 2. IPS tends to generate relatively concise responses.
> 3. With more complex previous contexts, we observed that IPS does not prioritize shortening the length of response. IPS can generate responses that are more in line with the situation based on the characteristics of the conversation.
>
> We will attach corresponding instances and the cosine similarity heatmap of responses generated by different decoding methods to show the effectiveness of IPS in our final version,  offering readers a better understanding of isotropy and proximity.
>
> ---
>
> **Concern:** The agreement of informativeness on two datasets
>
> **Response:** We sincerely appreciate your feedback. As indicated in human evaluation instructions, *informativeness measures whether the generated text has diverse, informative, novel, or logically related content*. So, it is essentially a **more subjective** evaluation metric than fluency, coherence, and semantic coverage. Additionally, the annotators we recruited for human evaluation are all native Chinese speakers, thus causing a higher disagreement on the Dailydialog dataset which is in English.
>
> ---
>
> **Concern:** Title of the paper
>
> **Response:** Thank you for your suggestion regarding the article title. We will change it to "Fine-grained Conversational Decoding via Proximal and Isotropic search" in the revision.
>
> ---
>
> Thank you again for your valuable suggestions to help us improve the quality of the work!

---

### Meta-Review · Area_Chair_Fg8g · 2023-09-19

**Recommendation:** 5

**Metareview:**

This paper proposes a new LLM decoding method called isotropic and proximal search based on locality and isotropy with a parameter to interpolate them. The reviewers appreciate the novelty and performance of the new method as well as the clarity of presentation. They raise a few concerns regarding evaluation, such as statistical significance and ablations, . The authors address all major concerns, conducting new experiments where needed. For these reasons I believe this paper should be accepted.

---

### Decision · Program_Chairs · 2023-10-07

**Decision:**

Accept-Main

**Comment:**

This paper proposes a new LLM decoding method called isotropic and proximal search based on locality and isotropy with a parameter to interpolate them. The reviewers appreciate the novelty and performance of the new method as well as the clarity of presentation. They raise a few concerns regarding evaluation, such as statistical significance and ablations, . The authors address all major concerns, conducting new experiments where needed. For these reasons I believe this paper should be accepted.